# Bromelain, a Group of Pineapple Proteolytic Complex Enzymes (*Ananas comosus*) and Their Possible Therapeutic and Clinical Effects. A Summary

**DOI:** 10.3390/foods10102249

**Published:** 2021-09-23

**Authors:** Carolina Varilla, Massimo Marcone, Lisete Paiva, Jose Baptista

**Affiliations:** 1Department of Food Science, University of Guelph, Guelph, ON N1G 2W1, Canada; mmarcone@uoguelph.ca; 2Department of Science of Physics, Chemistry and Engineering (DCFCE), Institute for Investigation and Technology of Agronomy and Environment (IITAA), University of Azores, 9500-321 Azores, Portugal; lisete.s.paiva@uac.pt (L.P.); jose.ab.baptista@uac.pt (J.B.)

**Keywords:** *Ananas comosus*, bromelain, proteinases, pineapple plant, phytotherapeutic applications, anti-inflammatory effect, clinical use of bromelain

## Abstract

Bromelain is a complex combination of multiple endopeptidases of thiol and other compounds derived from the pineapple fruit, stem and/or root. Fruit bromelain and stem bromelain are produced completely distinctly and comprise unique compounds of enzymes, and the descriptor “Bromelain” originally referred in actuality to stem bromelain. Due to the efficacy of oral administration in the body, as a safe phytotherapeutic medication, bromelain was commonly suited for patients due to lack of compromise in its peptidase efficacy and the absence of undesired side effects. Various in vivo and in vitro studies have shown that they are anti-edematous, anti-inflammatory, anti-cancerous, anti-thrombotic, fibrinolytic, and facilitate the death of apoptotic cells. The pharmacological properties of bromelain are, in part, related to its arachidonate cascade modulation, inhibition of platelet aggregation, such as interference with malignant cell growth; anti-inflammatory action; fibrinolytic activity; skin debridement properties, and reduction of the severe effects of SARS-Cov-2. In this paper, we concentrated primarily on the potential of bromelain’s important characteristics and meditative and therapeutic effects, along with the possible mechanism of action.

## 1. Introduction

Bromelain is the general term used for a group of proteolytic enzymes that are commonly associated with more specific endopeptidases present in the tissue of the Bromeliaceae plant family [1,2]. The most prominent plant belonging to this family is the pineapple (*Ananas comosus*). Bromelain was first identified in 1876 [3], but it was not fully isolated, purified and characterized until much later [4,5,6]. Although the pineapple plant contains a small amount of other proteolytic enzymes, such as ananain and comosain [7], bromelain remains the primary and most extensively explored among these proteolytic enzymes [8]. Physiologically, bromelain is present abundantly in both the stem and fruit of pineapple plants, with Heinecke first finding in 1957 [9] that there was considerably more bromelain in the pineapple stem than in the actual fruit. Later studies have shown that it can also be extracted in small amounts from other segments of the palm tree, such as the root system, the leaflet, and the rind [10,11,12,13,14,15].

The Enzyme Commission (EC) number 3.4.22.33 was given to bromelain isolated from the pineapple fruit and is referred to as fruit bromelain (FBM), whereas bromelain isolated from the pineapple stem is referred to as stem bromelain (SBM) and was assigned EC number 3.4.22.32 [16]. The higher overall SBM content in the plant has resulted in commercial development of the enzyme and, more recently, in its use as a potent phytomedical compound [15]. In turn, researchers have concentrated on the use of waste generated from pineapples as an economically feasible source of bromelain [4,17].

## 2. Physiochemical Properties and Pharmacokinetics of Bromelain

### 2.1. Structural Chemistry of Bromelain

Both stem and fruit bromelain are isolated as glycosylated, monomeric single proteins [2] with an isoelectric point (pI) value of 9.55 and optimum pH range of 6–7 [18]. The proteins also contain seven cysteine residues [19] and hence a total of three disulfide bonds. As such, they can also be further classified as cysteine proteinases, belonging to the common group of sulfhydryl proteolytic enzymes. Bromelain is also considered a thiol endopeptidase (internal clearing enzyme) [20], with the activity of all sulfhydryl endopeptidases depending upon the thiol group (SH) of a cysteine residue [3,21]. Stem bromelain has a stable secondary structure and shows activity in the pH range 7–10, with its activity being irreversibly lost above pH 10 [22]. A typical heated gelatinous mass conformation of stem bromelain is obtained at a pH of 14 and may exhibit charge heterogeneity, generating several variants when separated chromatographically. Although a confirmed mechanism thatcan explain this charge heterogeneity is currently not known, it has been shown that these variants are all immunologically similar [23]. Finally, bromelain is reported to be stable for an extended time if stored under −20 °C [24].

### 2.2. Comparison of Stem and Fruit Bromelain

The protease activity of SBM is comparatively higher than FBM [21,25,26] even though it possesses inferior specificity for peptide bonds. The optimal pH for SBM has been reported to be about 7 in numerous studies, whilst its optimal conditions range for demonstrating development is 50–60 °C [16,21,25,26]. The optimal pH condition for FBM, on the other hand, is 3–8, whereas the average maximum temperature is very wide, 37–70 °C [27,28]. Molecular weight for SBM is 26–37 kDa [29], whereas for FBM it is 24.5–32 kDa [21,27], both of which undergo complex glycosylation based on factors such as enzyme availability, amino acid sequence, and protein conformation.

### 2.3. Methods for Bromelain Isolation and Purification

Bromelain is one of the few plant proteases that can be extracted from a variety of plant components, including the fruit pulp, stem, peel, and leaves. The concentration of bromelain is higher in the stem than in the fruit and, therefore, the stem is one of the most available and abundant sources of bromelain [3]. Other parts outside the stem have also been investigated for their presence of bromelain [30], including the peel, core, and crown. Given this vast array of potential enzyme sources from the pineapple plant, researchers have now focused their efforts on the search for alternative, more efficient methods to obtain purified bromelain in fewer and more economical steps [4,29]. Micropropagation processes [31], reverse micellar systems [4,29], membrane filtration [27], and aqueous two-phase extraction [32,33] are the most studied and promising extraction methods for bromelain. Moreover, scientists continue to utilize the advance recombinant DNA technology to yield large-scale production and purification of recombinant bromelain [34].

### 2.4. Absorption and Bioavailability of Bromelain

Bromelain is absorbed efficiently in the human body and thus has a high bioavailability in an in vitro study [35]. This occurs as a result of its ability to bind to the two principal antiproteinases in the blood, alpha 2-macroglobulin and alpha1-antichymotrypsin. Without any significant side effects, it has been recorded that almost 12 g/day of bromelain can be taken in an in vivo study [36]. In addition, bromelain is absorbed in its active form throughout the gastrointestinal tract, with approximately 40 percent of the total bromelain absorbed in its high molecular weight form from the intestine [37]. In terms of pharmacokinetics, the maximal level of bromelain in the blood was obtained an hour after oral dosing [35]. Furthermore, [36] found that bromelain retains its plasma proteolytic function. Based on this, a more recent in vitro test showed that after 4 h, approximately 30% of bromelain remained stable in artificial stomach juice and that after the same time, roughly 20% of bromelain also remained stable in artificial blood [35].

## 3. Use of Bromelain in the Medical and Cosmetic Industries

As a phytomedical compound, bromelain was shown to exhibit a number of beneficial uses [15]. Bromelain has been shown to represent a crucial role as an antiedematous, fibrinolytic, anticancer, anti-inflammatory, antibiotic, anticoagulant, and antithrombotic agent [16]. In addition, it may have beneficial analgesic-type properties that can help to heal and minimize post-surgical pain and swelling [38]. Furthermore, bromelain has been shown to improve recovery after tissue damage and boost drug absorption, particularly for antibiotics [39]. In addition to its medicinal uses, bromelain has been utilized in many other industries such as the food and brewery industries as well as the clothing, cosmetic, and pharmaceutical sectors [34,40,41]. 

The industrial production of bromelain has been greatly increased due to its aforementioned phytomedical properties [34]. Given the breadth of scientific and technological uses of bromelain, this has opened the door for better and more profitable markets for pineapple wastes [42]. Through applying traditional and new isolation methods, and controlling the pH, hydrophobicity, and temperature conditions, researchers worldwide have focused their attention not only in crude isolation of this important enzyme but in further purification for other purposes [34,39,40,41,43]. It is for this reason that research into recombinant DNA technology has been pursued and has potentially enabled large-scale recombinant bromelain production and purification for innovative technologies [42].

### 3.1. Clinical Application

A number of studies have demonstrated that bromelain has a number of important therapeutic and application uses, as briefly discussed previously [44,45,46]. Bromelain boosts bioactivity and decreases the adverse impacts of multiple antibiotics [16]. Bromelain also functions as an immunomodulator, being antimetastatic, anti-edematous, antithrombotic, and anti-inflammatory in in vivo studies [22,47]. Furthermore, it has also been utilized as a primary therapeutic treatment for skin ailments, such as burns, infections, and skin debridement, as well as in the preparation of vaccines and antitumor agents [46]. Bromelain is highly absorbable by the stomach when administered orally without losing its biologically active compounds [46]. Most of bromelain’s biochemical function has been shown not to be due to one particular proteolytic component; rather, there are probably multiple factors involved in the positive effects of bromelain [44,45].

### 3.2. Effects of Bromelain on Blood Coagulation and Fibrinolysis

Anticoagulants are the group of drugs which can decrease, or inhibit, the coagulation of blood. They are used to clinically manage thrombosis as well as in the management and treatment of heart disease, with thrombosis being a major risk factor for mortality. According to a new report published by Allied Market research, Anticoagulants Market by Drug Class, Route of administration, Application: Global Opportunity Analysis and Industry Forecast 2018–2025, the market was valued at 24,265 million dollars in 2017 and was expected to reach 43,427 million dollars by 2025 (Anticoagulants Market by Drug Class, 2018–2025). The incidence of venous thrombosis has been reported to be 3% in the general population and, in the US, 100,000–300,000 people are reported to die from venous thrombosis each year, with the number of hospital admissions exceeding 500,000 within the same timeframe. In Europe, 500,000 deaths are reported because of venous thrombosis each year [48]. In 2018, the worldwide demand for anti-thrombotic drugs reached USD $25.9 billion in market sales [48]. Bromelain has been reported to have anticoagulant effects and is available at an average price of 40–80 US dollars/kg, depending on the source and quality (Pharmacompass.com) [48].

Bromelain affects blood clotting through the fibrinolytic ability of the serum and prevents the synthesis of blood-clotting proteins such as fibrin [49]. Bromelain restricts the production of fibrin by reducing some of the clot formation process intermediates (factor X and prothrombin) and accelerating fibrogenesis [48]. A rat study by Taussig et al. (1988) [50] showed a dose-dependent relationship of its activity on blood clotting. Both prothrombin time (PT) and partial thromboplastin triggered time (APTT) are significantly extended as the concentration increases. In both in vitro and in vivo studies, bromelain has been shown to be an effective fibrinolytic agent because it promotes the conversion of plasminogen to plasmin, resulting in increased fibrinolysis due to fibrin degradation [50]. Plasma prekallikrein is a proenzyme that has to be converted into kallikrein, the active form of the enzyme, which plays a role in coagulation, and bradykinin is a compound released into the blood that causes smooth muscle contraction and blood vessel dilation, further increasing coagulation [51,52]. In view of this, it has been documented that bromelain reduces prekallikrein synthesis and thus reduces bradykinin growth at the site of infection, relieves pain and inflammation, and improves blood circulation at the damage tissue in rat studies [52]. In the future, a more in-depth investigation of bromelain’s capacity to reduce blood clotting would be highly beneficial.

### 3.3. Effects of Bromelain on Cardiovascular Disease

Heart disease is the number one cause of deaths worldwide, according to the World Health Organization (WHO), and more individuals die from this disease annually than almost any other illness. Statistics indicate that in 2016, 17.9 million individuals died from cardiovascular diseases, comprising 31% of all deaths worldwide. Eighty-five percent of these deaths are due to heart attack or stroke, with the reported age of those affected between the 30 and 70 years old. Data provided by International Medical Statistics (IMS) showed that global market sales of anti-thrombotic drugs amounted to USD $23.5 billion in 2013, accounting for 53.1 percent of cardiovascular disease drug sales and 2.7 percent of all global drug sales, considering that cardiovascular diseases are caused by heart and blood vessel disorders [48,53]. Importantly, bromelain is effective in the treatment of cardiovascular diseases and has a lower cost than any cardiovascular drug [48]. According to amazon.com, marketed bromelain supplements that contain 1200–1800 capsules per bottle at a dose of 500 mg, are available in the price range of 14–30 US dollars depending on the source and supplying company (Sigma-Aldrich is selling 10 g of purified bromelain for 89 €).

It is also relevant to note that bromelain’s anticancer property is also mainly attributed to its capability to inhibit blood platelet aggregation [9]. The blood coagulation factors, fibrin, and thrombus formation are important players in blood coagulation [52] and fibrinolysis, as well as the inhibition of thrombus formation, impedes blood clotting in blood. In fact, studies have reported that bromelain presents fibrinolytic activity as well as the potential to inhibit thrombus formation and reduce platelet aggregation [16]. Bromelain causes thrombus destruction, decreases red blood cell clumping, and decreases blood viscosity and thereby reduces the incidence of angina pectoris and transient cardiomyopathy attacks [43]. Thrombophlebitis is an inflammatory process that helps a blood clot to form and block one or more veins, especially in legs. Bromelain aims to reduce the possible complications of thrombophlebitis and facilitates its rehabilitation [16]. By reducing walking difficulties in patients and related inflammatory symptoms, including skin temperature, tenderness, inflammation, and pain, studies have indicated that bromelain is effective in treating acute thrombophlebitis.

Bromelain can also inhibit attacks of angina, resulting in the symptomatic relief in hypertension [43]. According to several in vivo studies, bromelain increases the efficiency of the heart, improves arterial flow, decreases arterial dissections, and increases angiogenesis [52,54]. Previous studies have shown that treatment with bromelain decreases apoptosis and endothelial cell damage in hepatic ischemia [53,55] and has also been reported to protect against muscle tissue hepatic encephalopathy injury [54,55,56,57]. It increases the permeability of the blood vessel wall that, as consequence, improves the uptake of oxygen, nutrients, and blood fluidity [56,58]. In experimental animals, bromelain was found to play an anti-hypertensive role for a prolonged period of administration indicating that bromelain supplements can, therefore, also reduce risk factors for cardiovascular diseases (CVD) [16].

### 3.4. Effects of Bromelain on Cancer Cells

Cancer is one of the world’s largest causes of mortality, mainly in poor countries, as per the WHO. Worldwide, 14.1 million new cases and 8.2 million cancer-related deaths occurred in 2012 (Cancer Statistics, 2018). Naturally, the disease also places a large economic strain on healthcare systems, and the estimated national expenditure for cancer care in the United States was $147.3 billion in 2017, with further economic consequences being observed in terms of loss of productivity related to morbidity and premature mortality. In the US, cancer healthcare expenditures in 2017 were reported at US$161.2 billion; mortality productivity loss, US$30.3 billion; and early death, US$150.7 billion. Approximately 1.8 percent of the gross domestic product (GDP) is reported to represent this financial cancer incidence in the US (The Economic Burden of Cancer) [59].

Cell growth and proliferation in normal cells are regulated, and cell cycle disparities can result in abandoned cell growth and transformation into cancer cells. There are various mechanisms inside cells to defend their DNA from harmful toxins and genomic instability [57,60]. Tumor cells frequently lose checkpoint controls, which are responsible for normal cell cycle regulation [47], and, as a result, the regulation of cell cycle progression is an important approach for cancer chemotherapy. Bromelain has frequently been shown to have anticancer effects by many researchers [47,57,58]. Bromelain has been shown to prevent nuclear factor αB (NF-αB) translocation by G2/M arrest and to contribute to cell death in human epidermoid cancer cells [47]. The disruption of normal apoptotic functions influences cell transformation and helps cancer cell development. Cell shrinkage, chromatin condensation, DNA fragmentation, and activation of caspases are involved in the apoptotic process [61]. Mitochondrial (intrinsic) or death receptor pathways (extrinsic) typically obtain apoptosis. Bromelain selectively induces apoptosis in cancer cells by upregulating the expression of p53, and the mitochondrial apoptotic pathway is triggered by increased Bax expression and discharge of cytochrome c [62].

Bromelain induced cell growth inhibition in mouse tumor cell lines and expressed the potential for Matrigel invasion [54] while growth of Kato-III cell lines in gastric carcinoma was also significantly reduced by the therapy. Furthermore, bromelain slows down the growth inhibitory response of MCF-7 cells in mammary carcinoma cells and stimulates the autophagy cycle while it promotes monocytic cytotoxicity in women with breast cancer when administered orally [58]. Bromelain has also demonstrated in vivo antitumor activity in cell lines such as P-388 leukemia, sarcoma (S-37), Ehrlich ascetic tumor, Lewis lung cancer, and mammalian adenocarcinoma ADC-755. In these studies, intraperitoneal administration of bromelain after 24 h of tumor cell inoculation promoted tumor regression [47]. 5-Flurouracil (5-FU) is a drug used to treat cancers of the breast, colon, stomach, rectum, and pancreas. Therefore, it is interesting to note that the antitumor interaction of stem bromelain was found to be greater than that of 5-FU; compared to untreated patients, their survival index was around 263 percent. When administered with 5-FU, bromelain greatly decreased the degree of lung metastases caused by LLC transplants. Bromelain’s antitumor activity against S-37 and EAT, which are cancer types vulnerable to immune response facilitators, and unimpacted tumor growth in the metastatic model suggest that antimetastatic action happens as a consequence of a mechanism independent of the primary antitumor effect [47]. Bromelain’s anticancer activity is specifically related to its effect on cancer cells and their microenvironment, and to the manipulation of the immune, inflammatory, and hemostatic systems [57]. As suggested by numerous studies, NF-3B, Cyclooxygenase-2 (COX-2), and Prostaglandin E2 (PGE2) are boosters of cancer development, and it has been observed that NF-3B signaling and overexpression play a significant role in so many other cancer types [61]. COX-2, a multiple target gene of NF-κB, helps in the conversion of arachidonic acid into PGE2 and leads to tumor angiogenesis and progression [62]. Proper inhibition of the action of NF-κB, COX-2, and PGE2 can be a possible treatment for carcinoma. In this regard, studies have shown that bromelain down-regulates the expression of NF-κB and COX-2 in mouse carcinomas and in tests of skin tumorigenesis [63]. Bromelain is able to exert this effect by inhibiting bacterial endotoxin (LPS)-induced NF-κB activity and can also block the expression of PGE2 and COX-2 that has been shown in human monocytic leukemia and murine microglial cell lines [64,65].

### 3.5. Antimicrobial Activity of Bromelain

Bromelain prevents the development of intestinal bacteria, such as Escherichia coli and Vibrio cholera, and can also inhibit Escherichia coli (ETEC) enterotoxigenic bacteria. Studies have shown that ETEC receptors can be temporarily inactivated in vivo and exert this effect by interacting with pathways of intestinal secretory signaling, e.g., adenosine 3: 5′-cyclic monophosphatase, guanosine 3′: 5′-cyclic monophosphatase, and calcium-dependent signaling cascades. It can also defend against diarrhea from Escherichia coli and, therefore, bromelain shows significant use as a prophylaxis against ETEC infection [66].

Antihelminthic activity against gastrointestinal nematodes, including Trichuris muris, Trichoderma viride, and Heligmosomoides polygyrus, has also been documented for bromelain [67]. As such, it can counter a number of effects related to intestinal pathogens. Not only this, but bromelain has a synergistic effect when administered alongside antibiotics, and these two mechanisms can be explored when aiming to further uncover the benefits of bromelain against specific infections. In addition, by inducing phagocytosis and the pulmonary rupture killing of Candida albicans [68], bromelain has been reported to act as an anti-fungal agent. Another skin condition where bromelain has efficacy is Pityriasis lichenoides chronica, which has been shown to completely cure the disease caused by Pityriasis lichenoides [69]. According to evidence regarding bromelain’s synergistic effects, certain antibiotics (such as amoxicillin) are found to increase in blood and urine levels in humans while they are taking bromelain [69,70]. In respiratory infections such as influenza virus, chronic bronchitis, staph infection, thrombophlebitis, cellulitis, pyelonephritis, and in perirectal and rectal ulcers, sinus infections, and bladder infections, the integration of bromelain and antibiotic treatment has been shown to be more effective than antibiotics alone [46]. A formulation of bromelain, trypsin enzyme, and flavonoid pigment has been prescribed as an immunological agent combined with antibiotics for infants with septic shock. In addition, Bromelain increased protein consumption in elderly patients with decreased protein absorption in conjunction with enzymes derived from Aspergillus Niger fungus [71].

### 3.6. Application of Bromelain in Unhealthy Tissue Burns

The removal of damaged skin tissue from wounds or second/third-degree burns is called debridement. Quick debridement and/or elimination of eschar is an important step in the treatment of intense partial and full-thickness burns [72]. It aims to reduce wound bioburden and enables the early healing of wounds by careful care or skin grafting [73]. Effective removal of the eschar within 72 h is thought to boost the outcome of burn wound treatment [74]. Thirty-five percent bromelain in a lipid-based drug delivery system, when used as a cream, helps in necrotic tissue debridement and accelerates the healing process due to the presence of escharase in bromelain [74,75]. Escharase is non-proteolytic and cannot hydrolyze regular protein substrates or the multiple substrates of glycosaminoglycans [16]. Due to the risk of surgical operations combined with the risk of repeated anesthesia and extreme bleeding [76,77], bromelain enzyme debridement is more successful than surgical debridement. In two separate experiments, two bromelain-based agents, Debriding Gel Dressing and Debrase Gel Dressing, were able to rapidly eliminate the dead tissue layer of the dermis while simultaneously protecting the unburned tissue in a porcine model [72,75]. As bromelain facilitates the debridement mechanism and provides improved, faster healing and efficient reepithelialization, bromelain has been suggested as valid approach for treating postoperative wounds and relieving discomfort, swelling, and other side effects.

### 3.7. Anti-Inflammatory Activity of Bromelain

Although bromelain has many therapeutic functions, the most prominent effect is its anti-inflammatory action. Bromelain was prescribed as an adjunctive treatment strategy for chronic inflammatory, malignant, and autoimmune disorders and was reported to increase the treatment efficiency in the diseases [16,78]. In vitro research shows that bromelain has the ability to modulate T cells, phagocytosis, and surface adhesion molecules to kill bacteria cells, causing peripheral blood mononuclear cells (PBMCs) to secrete interleukin (IL)-1β, IL-6, and tumor necrosis factor α (TNFα) [79,80,81,82,83]. Mast cell proteases assist in the treatment of asthma and allergic disorders by inhibiting the concerned proteins and globulins [79,84]. In the treatment of patients’ respiratory problems and hyperactivity disorders, bromelain, as a protease itself, is thought to produce similar effects to those found with other proteases [85].

An integral component of cancer-associated inflammation, cyclooxygenase-2 is involved in PGE2 synthesis. PGE2 is a pro-inflammatory lipid that serves as both an immunosuppressant and a promoter of tumor progression [63]. COX-2 converts arachidonic acid into PGE2 and facilitates tumor angiogenesis and cancer growth [86]. In murine microglial cells and human monocytic leukemia cell lines, bromelain can down-regulate expression levels of COX-2 and PGE-2 [83]. Bromelain stimulates inflammatory mediators, particularly IL-1β, IL-6, interferon (INF)-g and TNF-α in human peripheral blood cells (PBMC) [79,80,87]. In vivo, it increases T cell-dependent immunity while blocking T cell responses in vitro and can increase T cell antigen-independent binding to monocytes [88]. These findings showed that bromelain, in conjunction with the quick reaction to cellular stress, effectively stimulates a stronger immune system. Alternatively, bromelain decreases the release of IL-1β, IL-6, and TNF-α because immune cells have already been involved in inflammation-induced cytokine generation [54]. By blocking the T cell signal transduction pathway, bromelain can therefore inhibit the Raf-1/extracellular-regulated-kinase- (ERK-) 2 pathways [66]. Bromelain has been shown to produce both analgesic and anti-inflammatory effects in patients with rheumatoid arthritis when taken orally [89] and to induce increased expression of TGF-β, one of the key inflammatory regulators in osteomyelofibrosis and rheumatoid arthritis patients [89,90]. Many studies have also documented the immunosuppressive influence of bromelain [91]. In particular, in their study, Stopper et al. (2003) [92] found that the use of bromelain in inflammatory bowel disease can minimize INF-g and TNF-alpha expression. Bromelain also reduced the cell-damaging effect of advanced glycation products (AGEs) through degradation of their proteolytic receptors and, consequently, regulated the inflammation. Regarding its analgesic properties, bromelain administration prior to surgery will also promote more prompt dissipation of discomfort and pain in the post-operative period. These analgesic properties also mean that in women who underwent episiotomy, bromelain can be useful in reducing swelling, bleeding, and discomfort.

Bromelain stimulates phagocytic activity and increases the production of the stimulating factor IL-2, IL-6 granulocyte-macrophage-colony and decreases the activation of the T cell. It decreases most inflammatory mediators and has been shown to play a major role as an anti-inflammatory agent in different circumstances [46].

#### 3.7.1. Allergic Airway Disease (AAD)

When altering populations of CD4+ and CD8+T lymphocytes, bromelain can amplify the development of allergic airway disease (AAD). This progress in AAD suggests that the treatment of human asthma and hyperactivity disorders can be substantially affected by bromelain [85].

#### 3.7.2. Arthritis

In 2018, more than 23 million people lived with arthritis, WHO announced. Adjusted 2015 estimates, however, indicate that the prevalence of arthritis in the US, particularly amongst people younger than 65 years old, has also been substantially underestimated. According to the report, either a doctor diagnosed 92.1 million individuals with arthritis or joint symptoms were identified in accordance with the diagnosis of arthritis [93,94]. Approximately one in three (both men and women) had doctor-diagnosed arthritis and/or reported joint symptoms consistent with the diagnosis of arthritis [93,94] for people aged 18 to 64 years. In 2013, the national healthcare expenses for arthritis were $140 billion, contributing to $2117 in extra medical expenses per adult (The Cost of Arthritis in US Adults, 2020). The low cost and wide availability of bromelain make it an excellent treatment for arthritis-related rheumatoid arthritis. Bromelain acts as a strong pain reliever in this role and has a significant impact on certain pain messengers, such as bradykinin, which promotes smooth muscle contraction and blood vessel dilation [51].

Given that bromelain significantly decreased pain and stiffness in patients with osteoarthritis [95], it may be considered a risk-free alternative. Moreover, studies have shown that bromelain has a positive effect on arthritis (osteoarthritis and rheumatoid arthritis) [16] and increases the treatment effectiveness of degenerative joint pain problems when used in combination with other nutraceuticals, such as turmeric [96]. Given these properties, it is not surprising that bromelain has been used as a food supplement alternative to nonsteroidal anti-inflammatory drugs (NSAIDs) [97]. Bromelain has a strong effect on pain mediators such as bradykinin, which contributes to analgesic properties [98]. A clinical study found that bromelain was successful in patients with inflammation due to arthritis and resulted in a substantial or total decrease in swelling of the soft tissue [99].

In summation, experimental evidence demonstrates that bromelain is effective in reducing mucosal cytokine secretion in inflammatory bowel disease (IBD). Inflamed tissue in IBD secreted less granulocyte colony-stimulating factor (G-CSF), granulocyte-macrophage colony-stimulating factor (GM-CSF), interferon (IFN)-gamma, CCL4/macrophage inhibitory protein (MIP)-1beta, and tumor necrosis factor (TNF) after being treated with bromelain in vitro.

#### 3.7.3. Colonic Inflammation

Worldwide, there were 6.8 million (95 percent UI 6.4–7.3) cases of colonic inflammation in 2017 [100]. Hospitalization was a significant portion of the healthcare expenditures associated with this disease, accounting for 49% to 80% of the overall costs, whereas the expense of surgery represented 40–61% of all hospitalization costs [101]. Bromelain can reduce the severity of colonic inflammation [102,103] and bromelain’s anti-inflammatory function also results in proteolytic action by eradicating cell surface receptors that are implicated in leukocyte defects and activation [104]. Bromelain also differs in the release of other chemokines, thus decreasing the incidence and frequency of persistent colitis. Bromelain is considered to be an innovative treatment for intestinal inflammation disease [105]. Bromelain administration reduces a number of pro-inflammatory molecules such as INF-g INF-γ and colony-stimulating factor [105,106] in patients with inflammatory bowel diseases. Patients with ulcerative colitis have also shown a significant symptomatic improvement following application of bromelain [106], and it has also been stated that the release of some chemokines has been impaired, thus reducing the incidence and severity of colitis [107].

#### 3.7.4. Sinus Inflammation

In 2018, the number of adults with diagnosed sinusitis was 28.9 million, which equates to 11.6% of all adults in the US (Summary Health Statistics Tables for US, 2018). Moreover, the overall chronic rhino sinusitis (CRS) related health care costs had been reported to be in the range of US $3.9 billion to US $12.5 billion per year [108]. Bromelain has been documented to be beneficial in the treatment of sinusitis and can be used to manage the infection [46]. Once more, the low cost of bromelain makes it an attractive alternative to established medicinal treatments. In children with serious sinus infections, treatment with bromelain has been reported to minimize the duration of symptoms and facilitate complete recovery compared with normal treatment regimens [109]. Patients with sinusitis have reported full relief from breathing problems and nasal mucosal infection [110].

### 3.8. Toxicity Profile of Bromelain

Bromelain has been reported to have almost no toxic effects in preclinical and clinical studies. Taussig et al. (1988) [50] presented findings highlighting that bromelain administration is associated with very minimal toxic effects in mice, rats, and rabbits, with lethal doses (LD50) greater than 10 g/kg body weight. There was also no toxicity during a six-month medication period in preclinical dog research, raising the dosage of bromelain to 750 mg/kg for each daily dose. In addition, 1500 mg/kg per day did not cause any carcinogenesis or mutagenicity effects when administered to rats and did not have a negative impact on food consumption, growth development, spleen, kidney, or hematological specifications. Additionally, bromelain (3000 FIP unit/day) was applied to the human body throughout a period of 10 days in a study by Eckert et al. (1999) [58], and no substantial effect was observed in blood clotting parameters.

### 3.9. Effect of Bromelain on SARS-CoV-2

The global COVID-19 pandemic is an emerging respiratory illness due to severe coronavirus 2 acute respiratory syndrome (SARS-CoV-2) that has been a devastating contagious disease, affecting more than 84,400 million people and causing more than 1799 million deaths globally, as of 30 December 2020 (John Hopkins COVID-19 dashboard) [111,112]. Recent research has shown that SARS-CoV-2 ties to the binding site of the angiotensin-converting enzyme 2 (ACE2) and activates the initiation of three routes: inflammatory disease, clotting, and cascading bradykinin [111,112]. This activation leads to a variety of serious health problems, ranging from immunocompromised to acute respiratory syndrome and ultimately death. Due to their potential antioxidant and anti-inflammatory properties, some natural products have been shown to mitigate the harmful effects of COVID-19. The immunosuppressive effects of bromelain (from the pineapple stem) and curcumin (from turmeric) have been shown to limit the significant factors of COVID-19 pathogenesis [113]. These actions include the inhibition of signaling pathways and, subsequently, the depletion of inflammatory cytokines responsible in the production of inflammatory cells that are critical for the growth of SARS-CoV-2 syndrome [46,114]. Furthermore, bromelain decreases the enzyme responsible for prostanoid formation and thus decreases prostaglandins and thromboxane, affecting the swelling and clotting pathways. In contrast, it is well reported that curcumin presents antiviral action against multiple viruses [115]. In addition, curcumin prevents the reduction of inflammation by ACE modulating angiotensin II, although it still induces fibrogenesis and anticoagulation [46,116]. Orally administered bromelain, on the other hand, promotes the absorption of curcumin, increases its bioavailability and, with affective anti-inflammatory agents and anti-clotting factors, provides a good nutraceutical immune boost [117,118]. Bromelain also hydrolyzes Angiotensin-converting enzyme and decreases levels of kininogen and bradykinin according to Lotz-Winter et al. (1990) [49]. Based on the research by Sagar et al. (2020) [119], Bromelain can resist SARS-CoV-2 infection by targeting ACE-2 and TMPRSS2, as well as SARS-CoV-2 S-proteins.

## 4. Conclusions

Owing to its remarkable uses in numerous fields, bromelain is one of the most thoroughly researched proteolytic enzymes. As such, many studies have focused on the search for novel strategies that can yield the highest amount of purified bromelain in a high-throughput and economically viable manner. Research shows that the choice of processing for various isolation methods, such as enzyme-mediated activity, bromelain composition, or substrate purification, depends on the intended use of the final product. Given this significant factor, chromatography for ion exchange is an expensive method, and precipitation and aqueous two-phase extraction requires high salt levels, making it difficult to extract and make a full recovery. Conversely, membranes have been optimized by observing the effect of polysaccharides and other macromolecules on the decrease in flow, exceeding the disadvantage of clogging residues on membrane spores. Notwithstanding these considerations, the research presented in this review indicates that bromelain is a safe and versatile therapeutic agent and is being used for a variety of therapeutic purposes, e.g., bronchitis, sinusitis, arthritis, and inflammation. It has also been reported to have both anticancer and antimicrobial effects, which can be further exploited. Following oral administration, bromelain is thoroughly stored in the body and retains a healthy efficacy and safety even after extended usage. Bromelain has the potential to be an effective health supplement that can be used to potentially reduce and combat cancer, diabetes, and multiple types of heart disease. The next step is to explore possible applications of bromelain in recombinant DNA technology as a biochemical catalyst for recombinant bromelain producers.

## Data Availability

Not applicable.

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
