# Peer review of "Bromelain, a Group of Pineapple Proteolytic Complex Enzymes (Ananas comosus) and Their Possible Therapeutic and Clinical Effects. A Summary"

_foods, 2021, doi:10.3390/foods10102249_

Round 1

Reviewer 1 Report

In general, this review does not provide a well-organized summary of the available knowledge about Bromelain, and the information given is often not precise  and confounding.   Just few examples of the many sentences which are not clearly formulated and which should be revised:

Lines 52-52: Both stem and fruit bromelain are isolated as glycosylated, monomeric single proteins with a pI of 9.55 and A1%, 280 of 20.1.

Line 59: Stem bromelain …shows activity in the pH range 7-10 ; line 69: the optimum pH range for SBM is in the range of 6-7

Lines 94-95: In terms of its pharmacokinetics, an hour following oral  administration, the maximum level of bromelain was obtained in the blood.

Lines 95-9:  Furthermore, [37] found that bromelain retains its plasma proteolytic function. Based on this, a more recent in vitro test showed that after 4 hours, 3.66 mg/mL of bromelain remained stable in artificial stomach juice and that after the same time, 2.44 mg/mL of bromelain also remained stable in artificial blood [36].

Lines 126-128: Oral administration bromelain has been reported to be consumed by the stomach without having to lose its biologically active compounds [50]

Lines 425-427 : it has been shown that the immunosuppressive effects of bromelain (from the  pineapple stem) and curcumin (found in turmeric) inhibit the appropriate measures of  Covid-19 pathogenesis [116].

A part the vagueness of the “appropriate measures of Covid-19 pathogenesis”, the last sentence is also one of the many examples of uncorrect quotations: in fact, ref 116 (2005, well before Covid-19 !) only refer to curcumin.

Another example is the very last sentence, from which it might appears that Bromelain might “minimize” a list of  Covid-19 manifestations (coughing, fatigue, pain, and other more serious consequences of  infection, such as cytokine production, to respiratory injury ) while the quoted ref. 122 only shows the in vitro  effects of Bromelain in VeroE6 cells.

Actually, the great part of the quoted literature appears wrong.  At line 88 it is written for example, that “Bromelain is absorbed efficiently in the human body and thus has a high bioavailability  [36]” but the title of ref 36 (proceedings)  is “In vitro study of bromelain activity in artificial stomach juice and blood”.  Other examples, among many are the citations of ref 16 (line 46), ref 17 (line 48), ref 40 (line 102), ref 41 ( line 104).

There are statements with no references, as the following examples:

“Bromelain has been reported to have anticoagulant effects and it is available at an average price of 40-80 US dollars/ kg, depending on the source and quality (Pharmacompass.com).” (lines 144-145).

“Importantly, bromelain is effective in the treatment of cardiovascular diseases and has a lower cost than any cardiovascular drug.” Lines 171-172

Finally, it is often not clear if the literature data were obtained in vitro, in animal models or in humans.

Author Response

Detailed List of Changes made to “Bromelain, a group of pineapple proteolytic complex enzymes (Ananas comosus) and their possible therapeutic and clinical effects. A Summary” i.e., Manuscript # Foods-1348080.

Responses to Reviewer 1

The authors would like to thank this reviewer for their very insightful comments. We have made all the changes requested by the reviewer and feel that these changes have improved the quality of the manuscript.

In general, this review does not provide a well-organized summary of the available knowledge about Bromelain, and the information given is often not precise  and confounding.   Just few examples of the many sentences which are not clearly formulated and which should be revised:

Lines 52-52: Both stem and fruit bromelain are isolated as glycosylated, monomeric single proteins with a pI of 9.55 and A1%, 280 of 20.1.

Change made: Both stem and fruit bromelain are isolated as glycosylated, monomeric single proteins [2] with an isoelectric point (pI) value of 9.55 and optimum pH range of 6–7.

Line 59: Stem bromelain …shows activity in the pH range 7-10 ; line 69: the optimum pH range for SBM is in the range of 6-7

Change made: The optimal pH for SBM has been reported to be about 7 in numerous research.

Lines 94-95: In terms of its pharmacokinetics, an hour following oral  administration, the maximum level of bromelain was obtained in the blood.

Change made: In terms of pharmacokinetics, the maximal level of bromelain in the blood was obtained an hour after oral dosing [36].

Lines 95-9:  Furthermore, [37] found that bromelain retains its plasma proteolytic function. Based on this, a more recent in vitro test showed that after 4 hours, 3.66 mg/mL of bromelain remained stable in artificial stomach juice and that after the same time, 2.44 mg/mL of bromelain also remained stable in artificial blood [36].

Change made: Based on this, a more recent in vitro test showed that after 4 hours, approximately 30% of bromelain remained stable in artificial stomach juice and that after the same time, roughly 20% of bromelain also remained stable in artificial blood [36].

Lines 126-128: Oral administration bromelain has been reported to be consumed by the stomach without having to lose its biologically active compounds [50]

Change Made: Bromelain is highly absorbable by the stomach when administered orally without having to lose its biologically active compounds[47]. Most of bromelain's biochemical function has been shown not to be due to one particular proteolytic component; rather, there are probably multiple factors involved in the positive effects of bromelain [45,46].

Lines 425-427 : it has been shown that the immunosuppressive effects of bromelain (from the  pineapple stem) and curcumin (found in turmeric) inhibit the appropriate measures of  Covid-19 pathogenesis [116].

Change made: The immunosuppressive effects of bromelain (from the pineapple stem) and curcumin (from turmeric) have been shown to limit the significant factors of Covid-19 pathogenesis [114].

A part the vagueness of the “appropriate measures of Covid-19 pathogenesis”, the last sentence is also one of the many examples of uncorrect quotations: in fact, ref 116 (2005, well before Covid-19 !) only refer to curcumin.

Change made: The immunosuppressive effects of bromelain (from the pineapple stem) and curcumin (from turmeric) have been shown to limit the significant factors of Covid-19 pathogenesis[114].

Another example is the very last sentence, from which it might appears that Bromelain might “minimize” a list of  Covid-19 manifestations (coughing, fatigue, pain, and other more serious consequences of  infection, such as cytokine production, to respiratory injury ) while the quoted ref. 122 only shows the in vitro  effects of Bromelain in VeroE6 cells.

Change made: ALL references has been corrected

Change made: Based on the research by Sagar et al.,2020 [120], Bromelain can resist SARS-CoV-2 infection by targeting ACE-2 and TMPRSS2, as well as SARS-CoV-2 S-proteins.

Actually, the great part of the quoted literature appears wrong.  At line 88 it is written for example, that “Bromelain is absorbed efficiently in the human body and thus has a high bioavailability  [36]” but the title of ref 36 (proceedings)  is “In vitro study of bromelain activity in artificial stomach juice and blood”.  Other examples, among many are the citations of ref 16 (line 46), ref 17 (line 48), ref 40 (line 102), ref 41 ( line 104).

Change made: Based on this, a more recent in vitro test showed that after 4 hours, approximately 30% of bromelain remained stable in artificial stomach juice and that after the same time, roughly 20% of bromelain also remained stable in artificial blood [36]

[36] Shiew PS, Fang YL, Majid FA. In vitro study of bromelain activity in artificial stomach juice and blood. In Proceedings of the 3rd International Conference on Biotechnology for the Wellness Industry, PWTC 2010 Oct.

Change made: line 46 The Enzyme Commission (EC) number 3.4.22.33 was given to bromelain isolated from the pineapple fruit and is referred to as fruit bromelain (FBM), while bromelain isolated from the pineapple stem is referred to as stem bromelain (SBM) and was assigned EC number 3.4.22.32 [16].

 [16] Pavan R, Jain S, Shraddha, Kumar A. Properties and Therapeutic Application of Bromelain: A Review. Biotechnol Res Int. 2012; 2012(1): 1-6.

Change made: Line 48 The higher overall SBM content in the plant has resulted in commercial development of the enzyme and, more recently, in its use as a potent phytomedical compound [15].

[15] Hossain MF, Akhtar S, Anwar M. Nutritional value and medicinal benefits of pineapple. International Journal of Nutrition and Food Sciences. 2015 Jan; 4(1):84-8.

Change made: Line 99 As a phytomedical compound, bromelain was shown to exhibit a number of beneficial uses [15].

[15] Hossain MF, Akhtar S, Anwar M. Nutritional value and medicinal benefits of pineapple. International Journal of Nutrition and Food Sciences. 2015 Jan; 4(1):84-8.

Change made: Line 100 In fact, as an antiedematous, fibrinolytic, anticancer, anti-inflammatory, antibiotic, anticoagulant, and antithrombotic agent, bromelain has been shown to represent a crucial role [16].

[16] Pavan R, Jain S, Shraddha, Kumar A. Properties and Therapeutic Application of Bromelain: A Review. Biotechnol Res Int. 2012; 2012(1): 1-6.

Change made: There are statements with no references, as the following examples:“Bromelain has been reported to have anticoagulant effects and it is available at an average price of 40-80 US dollars/ kg, depending on the source and quality (Pharmacompass.com).” (lines 144-145).[49]

Change made: Bromelain restricts the production of fibrin by reducing some of the clot formation process intermediates (factor X and prothrombin) and accelerating fibrogenesis[49].

Change made: “Importantly, bromelain is effective in the treatment of cardiovascular diseases and has a lower cost than any cardiovascular drug.” Lines 171-172[49].

Finally, it is often not clear if the literature data were obtained in vitro, in animal models or in humans.

Change made:: Also addressed in some paragraphs.

Change made: Bromelain is absorbed efficiently in the human body and thus has a high bioavailability in an in vitro research [36].

Change made: Without any significant side effects, it has been recorded that almost 12 g/day of bromelain can be taken an in vivo study[37].

Change made: Bromelain also functions as an immunomodulator, being antimetastatic, anti-edematous, antithrombotic, and anti-inflammatory in an in vivo studies [,22, 48]. In view of this, it has been documented that bromelain reduces prekallikrein synthesis and thus reduces bradykinin growth at either the site of infection, relieves pain and inflammation and improves blood circulation at the damage tissue in rat studies[53]

Reviewer 2 Report

The review "Bromelain, a group of pineapple proteolytic complex enzymes (Ananas comosus) and their possible therapeutic and clinical effects. A Summary" by Carolina Varilla and colleagues is a very extensive and very well written review. I have only one suggestion, if authors can change amount of Bromelain remaining in artificial stomach juice and artificial blood to percent remaining that would be great (section 2.4, line 97)

Overall I am OK with review, here are some suggestion for authors

There are few discrepancies in sentence written in the review:

  1. Authors mention an activity range for SBM is 7-10 pH (section 2.1) however, in section 2.2 authors mention that optimum range for SBM is 6-7. Is it typological error or something that authors want to make a point?
  2. Section 2.4 lines 97-99, is it possible to convert the amount remaining in artificial stomach juice and artificial blood in percent remaining. Because the readers don’t know the initial amount that was used for the study. There is no need to write based on, as it is totally different study.
  3.  Please rephrase the line “In addition to this, it has been documented that bromelain enhances recovery following tissue damage and increases the drug absorption, in especially antibiotics” line 106-107.
  4. Please rephrase the line 126-130 in section 3.1.
  5. Please mention the name of antibiotics in line 274. “With regard to the synergistic effects of bromelain, blood and urine levels of certain antibiotics in humans have been reported to increase.
  6. Please add reference for Bromelain study (line 425-427)
  7. Please remove line 433-434  “In addition, a current innovative research has shown that by blocking the viral binding sites  and/or the cell ligands (ACE-2 receptors and/or spike protein) [115], bromelain may also inhibit SARS-CoV-2 entry into cells” it is irrelevant.
  8. Remove line 440-443.

Please check the reference once again. 

Author Response

Responses to Reviewer 2

The authors would like to thank this reviewer for their comments which we have taken into consideration below. We would like to thank the reviewer for their favourable review of the manuscript and kind words. We appreciate that the reviewer found the manuscript to be very extensive and very well written.

Comments and Suggestions for Authors

The review "Bromelain, a group of pineapple proteolytic complex enzymes (Ananas comosus) and their possible therapeutic and clinical effects. A Summary" by Carolina Varilla and colleagues is a very extensive and very well written review. I have only one suggestion, if authors can change amount of Bromelain remaining in artificial stomach juice and artificial blood to percent remaining that would be great (section 2.4, line 97)

 Overall I am OK with review, here are some suggestion for authors

There are few discrepancies in sentence written in the review:

  1. Authors mention an activity range for SBM is 7-10 pH (section 2.1) however, in section 2.2 authors mention that optimum range for SBM is 6-7. Is it typological error or something that authors want to make a point?

Change made:: The optimal pH for SBM has been reported to be about 7 in numerous research, whilst its optimal conditions range for demonstrating development is 50-60oC [21,25,26,27].

  1. Section 2.4 lines 97-99, is it possible to convert the amount remaining in artificial stomach juice and artificial blood in percent remaining. Because the readers don’t know the initial amount that was used for the study. There is no need to write based on, as it is totally different study.

Change made:: Based on this, a more recent in vitro test showed that after 4 hours, approximately 30% of bromelain remained stable in artificial stomach juice and that after the same time, roughly 20% of bromelain also remained stable in artificial blood [36]

  1.  Please rephrase the line “In addition to this, it has been documented that bromelain enhances recovery following tissue damage and increases the drug absorption, in especially antibiotics” line 106-107.

Change made:  Furthermore, bromelain has been shown to improve recovery after tissue damage and boost drug absorption, particularly for antibiotics

  1. Please rephrase the line 126-130 in section 3.1.

Change made: Bromelain is highly absorbable by the stomach when administered orally without having to lose its biologically active compounds[50]. Most of bromelain's biochemical function has been shown not to be due to one particular proteolytic component; rather, there are probably multiple factors involved in the positive effects of bromelain [48,49].

  1. Please mention the name of antibiotics in line 274. “With regard to the synergistic effects of bromelain, blood and urine levels of certain antibiotics in humans have been reported to increase.

Change made: According to evidence regarding bromelain's synergistic effects, certain antibiotics (such as amoxicillin) are found to increase in blood and urine levels in humans while taking bromelain[72,73]

  1. Please add reference for Bromelain study (line 425-427)

Change made: The global Covid-19 pandemic is an emerging respiratory illness due to severe coronavirus 2 acute respiratory syndrome (SARS-CoV-2) that has been a devastating contagious disease, affecting more than 84,400 million people and causing more than 1,799 million deaths globally, as of December 30, 2020 (John Hopkins Covid-19 dashboard) [114,115].

  1. Please remove line 433-434  “In addition, a current innovative research has shown that by blocking the viral binding sites  and/or the cell ligands (ACE-2 receptors and/or spike protein) [115], bromelain may also inhibit SARS-CoV-2 entry into cells” it is irrelevant.

Change made:: Removed as requested

  1. Remove line 440-443.

Change made:: I do not know exactly which needs to be deleted since the line order is not exactly what I have in the manuscript.

I did not remove this line.

  1. Please check the reference once again. 

Change made::  We rechecked all references.

Reviewer 3 Report

In the review : “Bromelain, a group of pineapple proteolytic  complex enzymes (Ananas comosus) and their

 possible therapeutic and clinical effects. A  Summary”  the authors explained the potential therapeutic application of bromelain and also its possible action mechanism.   

Overall, this work is interesting as it is a wide scientific overview, from production to proposed applications of this interesting enzyme from pinapple. The authors clearly explain the rational of the study and the different field of application of bromelain. However, we would like to invite the authors  to clarify some points:

  1. Lines 24-25 please improve the language - “potential” is used twice in the  sentence
  2. Line 65 please change into “-20 °C “, similarly the Celsius degree are not correctly written, following, in line 70 and 71, pelase check all the document.
  3. Line 72, “the For SBM, the molecular weight range is 26-37kDa [30], while for FBM[21,28] it is 72 known to be 24.5-32kDa, as both enzymes are monomeric in nature”.maybe better to clarify if the diverse MW is due to diverse glycosilation, or other issue (e.g. partial proteolitic activity)
  4. Line 85-“Moreover, scientists continue to study recombinant DNA technology 85 for bromelain extraction in order to achieve more novel strategies for future utilization [35].” The concept is not clearly exposed, generally the recombinant DNA technology is aiming to replace extractive procedures with biotechnological processes. The authors should rephrase, or better explain this point.
  5.  
  6. Lines 96-99. The authors could check if more recent references are available in literature. In addition, there are also in vivo studies?

  1. Lines 146-160. Also in this case the authors could check if more recent references are available in literature. Moreover, it could be interesting to explain if in addition to in vitro and in vivo studies, clinical trials have been conducted or if they are planned for the future in order to deeper explore the bromelain ability to affect blood clotting.

  1. Lines 179-180 some of these concepts are already presented in the previous paragrapf, please cancel

  1. Line 191-192, poor language is used , please rephrase completely. It has been stated that bromelain improves efficient improvement of the  heart, improves arterial flow and decreases arterial dissection size and angiogenesis level [56,58].

  1. Lines 296-301- The authors should clarify if the performance of bromelin in debridments are supported by clinical trials or because of the foreseen activity it can be expected to improve wound treatments

  1. (INF)-ÿ- to be replaced with INF)-g in many places

  1. Line 345- please clarify if the effect is detrimental or not, the sentence is actually confusing.

  1. Lines 365-375. The authors clearly explained the potential application of bromelain in the field osteoarthritic diseases as food supplements. In fact, bromelain has an anti-inflammatory effect and it is able to reduce the pain. However, it could be useful explain through which molecular pathway it acts and if it is able to modulate the expression of specific biomarkers related to the pathology (e.g. MMPs or pro-inflammatory cytokines).

  1. In the conclusion section, the authors could better explain the possible application of bromelain in the field of recombinant DNA thechnology perhaps with some examples.

Author Response

Responses to Reviewer 3

The Authors would like to thank this reviewer for their comments which we have taken into consideration below. We would like to thank this reviewer as we have for Reviewer #2 for their favourable review of the manuscript and kind words.

Comments and Suggestions for Authors

In the review : “Bromelain, a group of pineapple proteolytic  complex enzymes (Ananas comosus) and their possible therapeutic and clinical effects. A  Summary”  the authors explained the potential therapeutic application of bromelain and also its possible action mechanism.   

Overall, this work is interesting as it is a wide scientific overview, from production to proposed applications of this interesting enzyme from pinapple. The authors clearly explain the rational of the study and the different field of application of bromelain. However, we would like to invite the authors  to clarify some points:

 We thank the reviewer for their kind words about the breath of the paper and its importance.

  1. Lines 24-25 please improve the language - “potential” is used twice in the  sentence

Deleted potential.

Change made: In this paper, we concentrated primarily on the potential of bromelain's important characteristics and meditative and therapeutic effects, along with the possible mechanism of action.

  1. Line 65 please change into “-20 °C “, similarly the Celsius degree are not correctly written, following, in line 70 and 71, pelase check all the document.

Change made:: Corrected all oC in the manuscript.

  1. Line 72, “the For SBM, the molecular weight range is 26-37kDa [30], while for FBM[21,28] it is 72 known to be 24.5-32kDa, as both enzymes are monomeric in nature”.maybe better to clarify if the diverse MW is due to diverse glycosilation, or other issue (e.g. partial proteolitic activity)

Change made:: Molecular weight for SBM is 26-37kDa [30], while for FBM it is 24.5-32kDa [21,28], both of which undergo complex glycosylation based on factors such as enzyme availability, amino acid sequence, and protein conformation.

  1. Line 85-“Moreover, scientists continue to study recombinant DNA technology 85 for bromelain extraction in order to achieve more novel strategies for future utilization [35].” The concept is not clearly exposed, generally the recombinant DNA technology is aiming to replace extractive procedures with biotechnological processes. The authors should rephrase, or better explain this point.

Change made:: Moreover, scientists continue to utilize the advance recombinant DNA technology to yield large-scale production and purification of recombinant bromelain. [35]

  1. Lines 96-99. The authors could check if more recent references are available in literature. In addition, there are also in vivo studies?

Change made:: Did not add to this.  

  1. Lines 146-160. Also in this case the authors could check if more recent references are available in literature. Moreover, it could be interesting to explain if in addition to in vitro and in vivo studies, clinical trials have been conducted or if they are planned for the future in order to deeper explore the bromelain ability to affect blood clotting.

Change made: In the future, a more in-depth investigation of bromelain's capacity to reduce blood clotting would be highly beneficial. 

  1. Lines 179-180 some of these concepts are already presented in the previous paragrapf, please cancel

Change made:: We did not remove this because we did not see the duplication in lines 179-180 of the manuscript. 

  1. Line 191-192, poor language is used , please rephrase completely. It has been stated that bromelain improves efficient improvement of the  heart, improves arterial flow and decreases arterial dissection size and angiogenesis level [56,58].

Change made:: According to research, bromelain increases the efficiency of the heart, improves arterial flow, decreases arterial dissections, and increases angiogenesis 

  1. Lines 296-301- The authors should clarify if the performance of bromelin in debridments are supported by clinical trials or because of the foreseen activity it can be expected to improve wound treatments

Change made:: In two separate experiments, two bromelain-based agents, Debriding Gel Dressing and Debrase Gel Dressing, were able to rapidly eliminate the dead tissue layer of the dermis while protecting the unburned tissue in a porcine model simultaneously [78,79] 

  1. (INF)-ÿ- to be replaced with INF)-g in many places

Change made:: Replaced (INF)-ÿ to (INF)-g

  1. Line 345- please clarify if the effect is detrimental or not, the sentence is actually confusing.

Change made: The low cost and wide availability of bromelain make it an excellent treatment for arthritis-related rheumatoid arthritis 

  1. Lines 365-375. The authors clearly explained the potential application of bromelain in the field osteoarthritic diseases as food supplements. In fact, bromelain has an anti-inflammatory effect and it is able to reduce the pain. However, it could be useful explain through which molecular pathway it acts and if it is able to modulate the expression of specific biomarkers related to the pathology (e.g. MMPs or pro-inflammatory cytokines). 

Change made: In summation, experimental evidence demonstrates that bromelain is effective in reducing mucosal cytokine secretion in inflammatory bowel disease (IBD). Inflamed tissue in IBD secreted less granulocyte colony-stimulating factor (G-CSF), granulocyte-macrophage colony-stimulating factor (GM-CSF), interferon (IFN)-gamma, CCL4/macrophage inhibitory protein (MIP)-1beta, and tumor necrosis factor (TNF) after being treated with bromelain in vitro.

  1. In the conclusion section, the authors could better explain the possible application of bromelain in the field of recombinant DNA technology perhaps with some examples.

Change made: The next step is to explore possible applications of bromelain in recombinant DNA technology as a biochemical catalyst for recombinant bromelain producers.

Round 2

Reviewer 1 Report

The authors addressed all the points raised to the previous version and the manuscript has been sufficiently improved.